# Enhancing Document Understanding with Group Position Embedding: A Novel Approach to Incorporate Layout Information

**Yuke Zhu , Yue Zhang[*], Dongdong Liu[‡], Chi Xie, Zihua Xiong, Bo Zheng[†], Sheng Guo**
MYbank, Ant Group
`{felix.yk,guosheng.guosheng}@mybank.cn`

## Abstract

Recent advancements in document understanding have been dominated by Large Language Models (LLMs) and Large Multimodal Models. However, enabling LLMs to comprehend complex document layouts and structural information often necessitates intricate network modifications or costly pre-training, limiting their practical applicability. In this paper, we introduce Group Position Embedding (GPE), a novel and efficient technique to enhance the layout understanding capabilities of LLMs without architectural changes or additional pre-training. GPE achieves this by strategically grouping the attention heads and feeding each group with distinct positional embeddings, effectively encoding layout information relevant to document comprehension. This simple yet powerful method allows for effective integration of layout information within the existing LLM framework. We evaluate GPE against several competitive baselines across five mainstream document tasks. We also introduce a challenging benchmark called BLADE, specifically designed to assess the layout comprehension capabilities. Extensive experiments on both established benchmarks and BLADE confirm the efficacy of GPE in significantly advancing the state-of-the-art in document understanding. Our code is available at https://github.com/antgroup/GroupPositionEmbedding.git.

## 1 Introduction

Understanding complex document using AI has long been a popular topic in academia. Since the advent of Transformers (Vaswani et al., 2017), a series of pretrained models (Xu et al., 2020; 2021; Huang et al., 2022) based on the Transformer architecture have been proposed. For an extended period, these methods have been the mainstream solutions for document understanding. Recently, large language models(LLMs) like ChatGPT (OpenAI, 2022) have exhibited impressive capabilities across various tasks, particularly their Zero-Shot abilities, which have quickly established them as a new paradigm for future AI applications. In terms of document tasks, LLM-based methods (Perot et al., 2023; He et al., 2023; Zhang et al., 2023; Ye et al., 2023a; Bai et al., 2023) have also emerged as a hot research direction.

The approaches to document processing based on LLMs have evolved into several paradigms. The first approach (Liu et al., 2023b; Zhang et al., 2023; Ye et al., 2023a; Bai et al., 2023; Shi et al., 2023) directly processes document images. It first encodes the image using a visual encoder and then inputs the encoded representation into the LLM. Such methods typically require substantial pre-training data to enable the model to learn how to understand documents based on visual information. The second approach (Perot et al., 2023; Luo et al., 2024; Liao et al., 2024; Wang et al., 2023) utilizes OCR (Optical Character Recognition) models to first convert the document into text, then fed them to LLM. This method does not require the use of costly Vision-Language Pretraining, thereby reducing the challenge of direct document image understanding for LLM. However, it struggles with preserving the original layout and structure information of the documents. To address this shortfall, recent efforts (Luo et al., 2024; Wang et al., 2023; Liao et al., 2024) have employed large-scale

---

[*] Major experimental contribution [‡] Major dataset contribution [†] Corresponding author.

data pre-training to achieve this. In summary, when employing these methods to enable LLMs to understand complex documents, extensive pre-training is indispensable.

In this paper, we introduce a novel and efficient method for modeling document layout information based on LLMs, characterized by its consice design that eliminates the need for costly pre-training. We design a new positional embedding scheme for layout information to help the LLM understand layout nuances. Notably, this approach does not rely on complex visual encoders or introduce any additional learnable parameters. Instead, we incorporate layout information solely by modifying the existing position embedding of LLM. Specifically, we view the layout information as a multi-dimensional vector and use multiple sets of position embeddings to represent it. Inspired by the independence of attention heads in the multi-head attention mechanism, we partition the attention heads into groups as needed. Each group incorporates its corresponding positional information. We refer to this proposed positional embedding as Group Position Embedding (GPE). Its conceptual foundation and implementation are remarkably straightforward. In the context of document-specific tasks, we employ GPE to encode the coordinates of textual elements, effectively modeling layout information. The large language model, modified with GPE, does not incur any additional computational overhead or alter the original model architecture; thus, it can achieve the goal of understanding complex documents with minimal fine-tuning.

Some recent works (Luo et al., 2024; Wang et al., 2023; Liao et al., 2024; Fujitake, 2024; Lu et al., 2024a; Perot et al., 2023) also attempt to incorporate layout information into large language models. The simplest approach involves flattening the text along with its corresponding coordinate boxes (Perot et al., 2023) before inputting them into the model. However, this method not only increases the input length but also disrupts the original coherence of the text, and it does not necessarily guarantee that the model will learn the layout information embedded in the coordinate boxes. Alternatively, some methods treat coordinate box information as an additional modality (Wang et al., 2023; Fujitake, 2024; Liao et al., 2024), integrating it with the original text before feeding it to the model. The introduction of this new modality necessitates extensive pre-training to achieve modality alignment. Other approaches (Luo et al., 2024) have introduced more complex layout information encoding network, which increase computational load in order to inject layout information. In contrast, our method preserves the original input structure of the text and employs an extremely simple approach to model layout information.

For verification, we conducted experiments on several different base models and benchmarked our method on five mainstream documental tasks. Additionally, we introduced a more challenging Benchmark for Layout Analysis and Document Evaluation (BLADE). It is meticulously collected and designed to ensure that layout information is a critical factor for correctly answering questions. Through a series of experiments, we observed that our method effectively enables various LLMs to learn layout information and demonstrates highly competitive results compared to similar methods. In summary, the contributions of our method include the following points:

1. A universal and extremely simple positional embedding called Group Position Embedding, which helps LLMs understand document layout information.

2. A new and more challenging document evaluation benchmark, BLADE, designed to assess LLMs' comprehension of complex layouts.

3. Comprehensive experiments and valuable insights into model behavior.

## 2 RELATED WORK

**MLLMs for document understanding.** Recently, large language models represented by Chat-GPT(OpenAI, 2022) have demonstrated astonishing performance in text tasks, particularly in Zero-Shot tasks. To enable LLMs to understand more complex modal information, various Large Multi-modal Models (Liu et al., 2023a; Zhang et al., 2023; Ye et al., 2023b; Bai et al., 2023) have shown impressive performance in adapting to different modalities. In document-related tasks, multimodal LLMs can be categorized into two types. One type employs an end-to-end approach (Liu et al., 2023a; Zhang et al., 2023; Ye et al., 2023b; Bai et al., 2023), which directly accepts raw image inputs and processes document image inputs in an end-to-end manner. This approach requires the incorporation of an additional visual encoder and relies on costly pre-training processes to achieve fine-grained alignment between the image and text modalities. The other type (Luo et al., 2024;

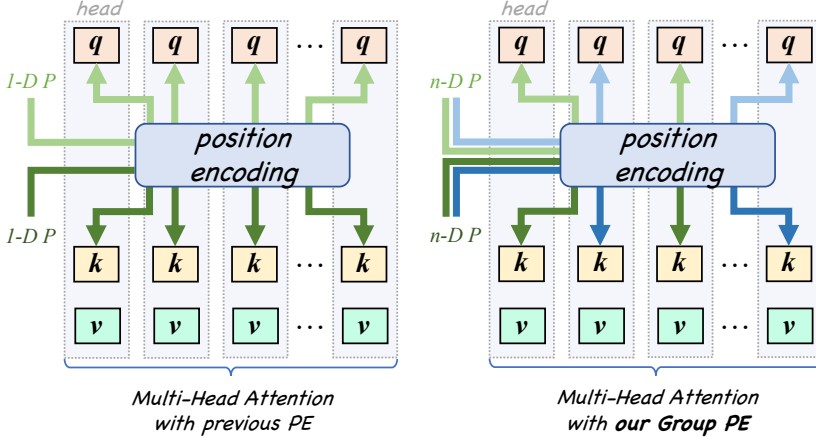

Figure 1: The concept of Group Position Embedding. Left: The standard multi-head attention, in which identical position embedding is applied to all heads. Right: Proposed group position embedding, in which positional information differs across head groups.

Wang et al., 2023; Liao et al., 2024; Lu et al., 2024a; Perot et al., 2023) utilizes traditional OCR models to first convert document images into text and text box information before using the LLM to process this information. This latter method was widely used before the advent of LLMs and has proven effective at learning complex document layout information. Exploring how to integrate LLMs with document layout information is a research direction worth delving into. Our proposed Group Position Embedding is based on the second approach.

**Layout-aware Position Embeddings.** Positional embedding is a common method used in structures like Transformers to capture sequence positional information. Earlier sinusoidal positional encodings (Vaswani et al., 2017) were frequently used in tasks such as language modeling and machine translation. The original positional embedding was mainly designed to represent the positional information of one-dimensional sequences, but many tasks require two-dimensional or multi-dimensional positional information. In image-related tasks, it is necessary to capture positional information in both width and height dimensions. A series of works exemplified by ViT (Dosovitskiy, 2020) achieve this by concatenating positional embeddings from two dimensions to represent 2D information. This concatenation method is also very common in detection tasks; for instance, in works like Deformable-DETR (Zhu et al., 2020), the positional embeddings of the bounding boxes are generated by encoding information for each dimension separately and then concatenating them together. In document tasks, LayoutLM (Xu et al., 2020), as an early pioneer, represents the spatial layout of documents by encoding the information from the corresponding bounding boxes for the text. It separately encodes each dimension and then sums the positional vectors of different dimensions at the input stage. Subsequent improvements (Xu et al., 2020; 2021; Huang et al., 2022; Li et al., 2021b;a;c; Appalaraju et al., 2021; Gu et al., 2021; Wang et al., 2022; Gu et al., 2022; Hong et al., 2022; Yu et al., 2023; Peng et al., 2022; Luo et al., 2023; Da et al., 2023) based on LayoutLM have continued to utilize similar coordinate encoding schemes. Compared with these approches, GPE is an entirely new method that can be combined with any positional encoding and quickly applied to models.

## 3 METHOD

In this section, we will first review the commonly used position embeddings in large language models. Later, we will illustrate the idea of Group Position Embedding(GPE) for LLMs and how to apply the grouping technique to the commonly used position embeddings. Lastly, focusing on the problem of document understanding, we will demonstrate how LLM can be facilitated to comprehend complex positional information.

### 3.1 REVIEW OF POSITION EMBEDDINGS

We review two representative position embeddings. The first is Sinusoidal Position Embedding, which serves as an exemplar of additive position embedding. The second is Rotary Position Embedding, representing the category of multiplicative position embedding.

**Sinusoidal Position Embeddings.** Sinusoidal Position Embeddings (Vaswani et al., 2017) are constant vectors that are added to token embeddings on input to the first layer of the transformer. They are pre-generated using sinusoidal function and selected based on the token index during model inference. For a given index $m$, the calculation process is as follows

$$PE(m)_{2i+1} = \cos(m/100000^{2i/d}), PE(m)_{2i} = \sin(m/10000^{2i/d}) \tag{1}$$

where $PE(m)$ denotes the Sinusoidal Embedding function. $2i$ or $2i+1$ denotes the index of the embedding channels and $d$ is the dimension of the embedding. The calculated position embedding is then added to the input embedding $\mathbf{x}$ as $PE(\mathbf{x}, m) = \mathbf{x} + PE(m)$.

**Rotary Position Embeddings (RoPE).** Rotary position Embedding (Su et al., 2024) is the most widely used position embedding in current LLMs. It ingeniously combines relative position encoding with absolute position encoding and excels in length normalization. Given an input $\mathbf{x}$ and its corresponding position $m$, Rotary position Embedding (RoPE) is added as following steps. First, the embedding $\mathbf{x}$ is viewed as a set of complex numbers by pairing consecutive elements of the vector: $\mathbf{x} = (x_1 + ix_2, x_3 + ix_4, ..., x_{d-1} + ix_d)$. It can also be denoted as $\mathbf{x} = \sum_{j=1}^{d/2}(x_j + ix_{j+1})\vec{\mathbf{e}_j}$ where $\vec{\mathbf{e}_j}$ is the $j$-th union direction vector. Then the RoPE for $\mathbf{x}$ is applied by

$$f(\mathbf{x}, m) = \sum_{j=1}^{d/2} x_j e^{im\theta_j} \vec{\mathbf{e}_j} \tag{2}$$

Based on RoPE, given query and key vector $\mathbf{q}, \mathbf{k}$, the attention score with position embedding is calculated as $< f(\mathbf{q}, m), f(\mathbf{k}, n) > = \sum_{j=1}^{d/2} q_j k_j e^{i(m-n)\theta_j}$.

### 3.2 GROUP POSITION EMBEDDING

Group Position Embedding is proposed in order to feed multi-dimensional position information to transformers. As illustrated in Figure-1, in the conventional multi-head attention mechanism, all heads share a common set of position embeddings. Following the intent of multi-head attention, GPE forces different heads to attend to different positional information. Given a multi-dimensional position $\mathbf{P}$ with dimension $l$ and the number of attention heads $h$. Denote $x$ as the input embedding The computation of GPE unfolds as two steps. First, group attention heads to match the dimension of position $\mathbf{P}$. We denote $Gr(i)$ as the group mapping function, where $i \in [1, 2, ..., h]$ and $Gr(i) \in [1, 2, ..., l]$. Second, using the selected position embedding to apply the position information to each head. For Sinusoidal Position Embedding, we use $GroupPE(\mathbf{x}, \mathbf{P})$ as the position embedding function. It writes as

$$GroupPE(\mathbf{x}, \mathbf{P}, i) = MLP_i(\mathbf{x}) + PE(\mathbf{P}_{Gr(i)}) \tag{3}$$

where $i$ is the head index and $MLP_i(\mathbf{x})$ projects the input embedding to $i$-th query vector or key vector. For Rotary Position Embedding, we use $GroupRoPE(\mathbf{x}, \mathbf{P})$ as the position embedding function. It writes as

$$GroupRoPE(\mathbf{x}, \mathbf{P}, i) = f(MLP_i(\mathbf{x}), \mathbf{P}_{Gr(i)}) \tag{4}$$

Grouping in a multi-head manner is a general technique and can be applied to any position embeddings. It ensures that positional information from different dimensions holds an entirely equal status, collectively participating in the inference process of the large language model. The grouping technique is completely compatible with the original one-dimensional position encoding. For a pure text input, we can just design $Gr(i)$ as an identity mapping, indicating that all heads being mapped to identical position index.

### 3.3 GROUP POSITION EMBEDDING FOR DOCUMENT UNDERSTANDING

In this section, we introduce the application of the GPE to the task of understanding complex documents. Conventionally, to enable LLMs to comprehend a typical document, Optical Character

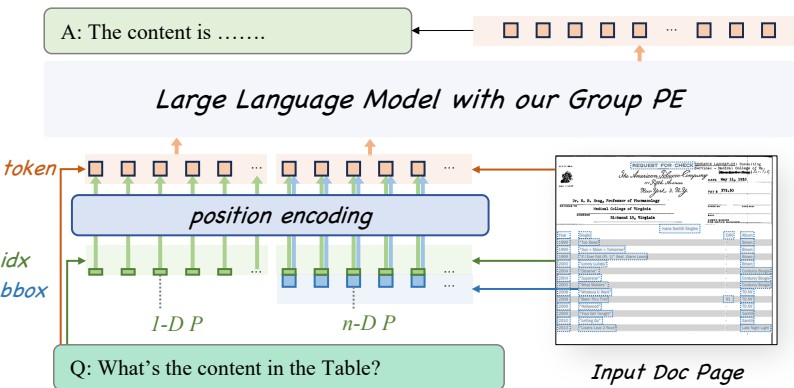

Figure 2: Group Position Embeddings for document understanding. Each token is equipped with a N-D positional information, which includes its reading order and corresponding bounding box.

Recognition (OCR) techniques, among others, are first employed to extract text boxes and textual information from the document. This is followed by ordering this information according to certain rules before feeding it into the LLM for processing. However, in more generic document tasks, obtaining a one-dimensional sorting can be immensely challenging due to elements such as tables and intricate layouts, necessitating the design of an exceedingly complex processing system. With GPE, we don't need to restore the exact positional order for each token; instead, we can simply assign ordering information using some straightforward strategies. For instance, we could adopt a global left-to-right sequence to index tokens, or directly assign them indices corresponding to their positions within local bounding boxes. Both approaches have their pros and cons in practice, which we will delve into further in subsequent discussions.

Specifically, given a token embedding $\mathbf{x}$, we consider its positional information from two perspectives. Firstly, in terms of reading order, $\mathbf{x}$ corresponds to a position id $m$. Secondly, we consider the spatial coordinates $\mathbf{B}$ of the bounding box where $\mathbf{x}$ resides. Without loss of generality, we utilize the coordinates of the top-left and bottom-right corners of the bounding box to denote its spatial position, i.e., $\mathbf{B} = [x0, y0, x1, y1]$. In practice, when handling spatial positions, we normalize the coordinates $\mathbf{B}$, resulting in $\bar{x}_i = \lambda * \frac{x_i - min(x)}{max(x) - min(x)}, \bar{y}_i = \lambda * \frac{y_i - min(y)}{max(y) - min(y)}$, where $\lambda$ is the scaling factor. It is set to 1000 in our implementation. Consequently, the positional information to be encoded for token embedding $\mathbf{x}$ can be represented as $\mathbf{P} = [m, \mathbf{B}]$. For the grouping function, we map the majority of heads to the layout part, each dimension using equal heads, the remaining for the reading order. Assuming the total number of heads is 32, the $Gr(i)$ is defined as

$$Gr(i) = \begin{cases} 1 & \text{if } 0 < i \leq 4, \\ \lceil (i-4)/7 + 1 \rceil & \text{if } 5 \leq i \leq 32, \end{cases} \tag{5}$$

For the instruction part, only its one-dimensional positional information is considered, and this portion is encoded following the conventional manner in which LLMs process text sequences. An illustration of our method for imposing positional encoding on inputs of complex documents is provided in Figure-2, using a document example.

Notably, there are various encoding schemes for the layout information of documents. For instance, we could choose to encode all four corner points of the bounding box to accommodate more complex scenarios. It will be discussed in experiments.

## 4  BLADE: A NEW BENCHMARK FOR COMPLEX LAYOUT ANALYSIS

We construct a more challenging Benchmark for Layout Analysis and Document Evaluation, named BLADE, in response to the saturation trend seen in current common document evaluation benchmarks. BLADE focuses on complex layout issues, with the majority of documents manually curated to ensure richness in layout information, thereby increasing the difficulty of assessment. Furthermore, the questions formulated based on these documents have been meticulously designed to ensure

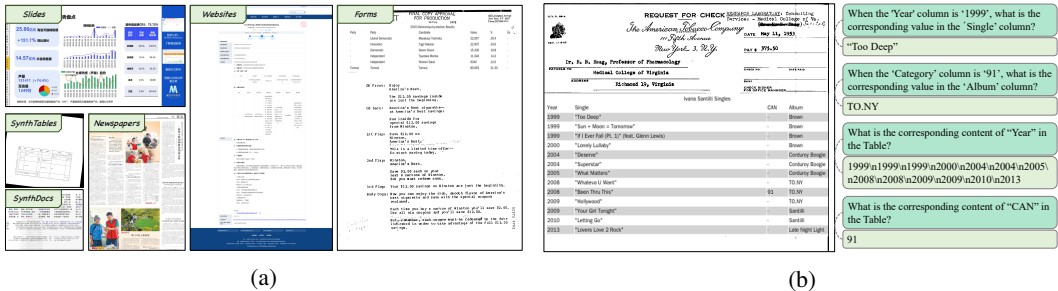

|  |  |
|---|---|
| (a) | (b) |

Figure 3: (a) Samples from 6 scenarios of BLADE. (b) An example from Forms, in which several questions are listed. Answering these questions necessitates the structural info of each element, making the dataset rather challenging.

that correct answers can only be inferred with the aid of spatial layout information. Examples from some of these evaluation sets are illustrated in Figure-3a and Figure-3b. It can be seen that they are quite challenging for large language models that lack spatial layout understanding.

The data in BLADE can be divided into two major categories. The first category comprises real-world data, including newspapers, web pages, slides, and various forms. The second category consists of synthesized data generated through rendering, involving synthetic documents and tables. For the real data, we primarily rely on manual annotation methods combined with the utilization of large language models to create question-answer pairs. In contrast, for the synthetic data, question-answer pairs are constructed using template questions. For details of BLADE, refer to the appendix.

## 5 EXPERIMENT

### 5.1 DATASETS

For the training process of LLM with GPE, we collect several publicly available document understanding datasets. They includes two synthetic datasets: Document Dense Description (DDD) and Layout-aware SFT used in (Luo et al., 2024) and four VisualQA datasets:DocVQA(Mathew et al., 2021), InfoVQA (Mathew et al., 2022), ChartQA(Masry et al., 2022), VisualMRC (Tanaka et al., 2021) and three KIE datasets:SROIE (Huang et al., 2019), CORD(Park et al., 2019), and FUNSD (Jaume et al., 2019). Besides, we also incorporate a proportion of instruction data for fine-tuning large models to combat the decline in the model's inherent capabilities after being trained on document data. For details of the training set, refer to the appendix.

### 5.2 IMPLEMENTATION DETAILS

For implementation, we selected several representative base models to apply GPE, including the Llama2-7B (Touvron et al., 2023), qwen2-7B (Yang et al., 2024), and ChatGLM-6B (GLM et al., 2024). Although these models are all based on the transformer architecture, they exhibit variations in certain details. For instance, qwen2 (Yang et al., 2024) uses group query attention instead of standard multi-head attention. The ChatGLM (GLM et al., 2024) adopts a prefix structure and utilize a bidirectional attention mechanism for the historical tokens. Since these structural details are related to GPE, we select them to validate our approach, thereby demonstrating its generality. For the training recipe, we use a learning rate of 1e-5 with 750 warmup steps and constant learning rate schedule, and Adam optimizer (Kingma, 2014) with beta1=0.9, beta2=0.99 and weight decay of 0.1. The maximum sequence length is set to 4096 for all base models during the entire training process. The training also involves the bf16 precision on 8 80GB A100 GPUs using the deepspeed training framework (Rasley et al., 2020) to complete one epoch.

### 5.3 RESULTS

In this section, we compare our method based on GPE with several document-oriented approaches across several commonly used benchmarks. We use ANLS(Mathew et al., 2021) as the evaluation

Table 1: Comparison with related methods on common document benchmarks. Note LLMs and ours methods are trained under identical setting. MLLMs' results are implemented using their provided weights. Results of LLMs with layout are copied from their papers. We use ANLS as the default evaluation metric for all benchmarks except VisualMRC, for which we use CIDEr.

| | Method | data amount | Document VQA | | VIE | | |
|---|---|---|---|---|---|---|---|
| | | | DocVQA | VisualMRC | FUNSD | CORD | SROIE |
| LLM | ChatGLM-6B | | 26.1 | 131.3 | 25.8 [*] | 68.0 [*] | 65.1 [*] |
| *Plain Text* | Llama-2-7B-Chat | 0.5M | 66.6 | 335.4 | 62.7 [*] | 75.6 [*] | 88.8 [*] |
| | Qwen2-7B | | 71.8 | **367.4** | 78.1 [*] | 83.4 [*] | 97.7 [*] |
| MLLM | LLaVAR-7B[†] | 1.2M | 11.6 | - | 1.7 | 13.6 | 2.4 |
| | LLaVA-1.5-7B[†] | 1.2M | 13.3 | - | 1.9 | 18.1 | 3.8 |
| | Qwen-VL-7B[‡] | 1.4B | 65.1 | - | 47.1 | 30.0 | 58.6 |
| | TextMonkey+[‡] | 2.6M | 66.7 | - | 42.9 | - | 46.2 |
| LLM | DocLayLLM(*with image*)[†] | 3.1M | 72.8 | 310.6 | 80.7 | 79.4 | 84.4 |
| | LayTextLLM-Llama2-7B | 6M | 77.2 | 277.8 | 81.0 [*] | 82.5 [*] | 96.1 [*] |
| *with Layout* | LayoutLLM-Vicuna-7B[†] | 6M | 74.3 | - | 78.7 [*] | 62.2 [*] | 71.0 [*] |
| | DocLLM-Llama2-7B | 3.8M | 69.5 | 264.1 | 51.8 | 67.4 | 91.9 |
| **ours** | **GPE-GLM-7B** | | 57.9 | 129.3 | 69.6 [*] | 83.0 [*] | 94.8 [*] |
| | **GPE-Llama2-7B** | 0.5M | 77.2 | 318.9 | 77.6 [*] | 84.7 [*] | 97.9 [*] |
| | **GPE-Qwen2-7B** | | **78.1** | 344.0 | **82.6**[*] | **86.9**[*] | **97.8**[*] |

[†] fully zero-shot setting, [‡] only zero-shot on VIE. *test set from (Luo et al., 2024)

metric for all benchmarks except VisualMRC, which we use CIDEr(Vedantam et al., 2015) as the metric. The comparative methods can be broadly categorized into three types: the first comprising pure LLMs, for which we selected ChatGLM6B(GLM et al., 2024), Llama2-7B(Touvron et al., 2023), and Qwen2-7B(Yang et al., 2024); the second type involves end-to-end Large Multimodal Models, including LLaVAR-7B(Zhang et al., 2023), LLaVA-1.5-7B(Liu et al., 2023a), Qwen-VL-7B(Bai et al., 2023) and TextMonkey+ (Liu et al., 2024); and the third category consists of LLMs that model layout information, exemplified by LayoutLLM(Luo et al., 2024), DocLLM(Wang et al., 2023), DocLayLLM(Liao et al., 2024) and LayTextLLM (Lu et al., 2024b). We implemented GPE on the above chosen pure LLMs, yielding GPE-GLM-6B, GPE-Llama2-7B, and GPE-Qwen2-7B. Note that for pure LLMs, they are tuned under the identical setting of GPE. Considering that these methods are based on different base models, trained with varying datasets and strategies, and some without publicly available model weights, the comparison is not entirely equitable. Nonetheless, valuable insights emerge from the comparative results presented in Table-1.

Our findings reveal that, after integrating GPE, LLMs exhibit notable improvements across various benchmarks compared to their original counterparts, affirming the efficacy of our proposed approach. Remarkably, GPE achieves highly competitive performance with significantly less training data compared to other methods. Additionally, we observe that end-to-end approaches fare less impressively on these benchmarks, as these tasks are inherently more challenging for such models, requiring them to also extract textual information from images. A noteworthy observation is the exceptional performance of the Qwen2-7B, particularly in VisualMRC, outperforming all other methods. This suggests that most of the benchmarks assess a model's text comprehension capability. Our analysis of these benchmarks(refer to the appendix) also indicates that scores on these benchmarks do not readily measure a model's ability to understand complex layout information. It is under this context that we introduce a new benchmark, BLADE, designed to address these limitations. Another observation is that GPE gets inferior performance compared with Qwen2-7B, it means that GPE may slightly influence the LLM's text comprehension capability. This phenomenon is further disccused in the appendix.

## 5.4 COMPARISON

To ensure a fair comparison between GPE and similar methods, we conduct a comparative experiment in this section using the same base model, dataset, and training strategy. We compare GPE with three representative methods. 1. Text Box (Perot et al., 2023): This approach directly represents bounding boxes as text, inserts the coordinate text into the original information text, and feeds the resulting new text sequence into the large language model as input. 2. Add Box Embedding (Xu

Table 2: Comparison with other layout-aware methods on common document benchmarks. Methods in this table are trained under identical setting. Qwen2-7B is selected as the base model. We use ANLS as the evaluation metric for all benchmarks except VisualMRC, for which we use CIDEr.

| Method | Document VQA | | VIE | | |
|---|---|---|---|---|---|
| | DocVQA | VisualMRC | FUNSD | CORD | SROIE |
| Raw Text | 71.8 | **367.4** | 78.1 | 83.4 | 97.7 |
| Text Box | 71.5 | 357.5 | 78.4 | 81.5 | **98.0** |
| Insert Box Embedding | 67.8 | 301.4 | 71.2 | 63.8 | 90.3 |
| Add Box Embedding | 71.5 | 350.1 | 78.4 | 79.5 | 87.9 |
| GPE | **78.1** | 344.0 | **82.6** | **86.9** | 97.8 |

Table 3: Comparison with other layout-aware methods on BLADE. Methods in this table are trained under identical setting. Qwen2-7B is selected as the base model.

| Method | SynthTables | Forms | Slides | Websites | SynthDocs | Newspapers |
|---|---|---|---|---|---|---|
| Raw Text | 50.3 | 33.0 | 42.2 | 46.1 | 53.3 | 43.8 |
| Text Box | 25.0 | 24.9 | 25.9 | 39.6 | 29.0 | 31.9 |
| Insert Box Embedding | 43.4 | 30.9 | 36.5 | 41.6 | 31.7 | 31.8 |
| Add Box Embedding | 51.4 | 32.8 | 43.2 | 50.3 | 57.6 | 45.1 |
| GPE-Sinusoidal | 67.2 | 45.6 | 48.3 | 66.5 | 55.2 | 48.7 |
| GPE-RoPE | **77.3** | **57.2** | **64.5** | **79.3** | **62.2** | **52.2** |

et al., 2020): A commonly employed strategy in conventional methods, it encodes bounding boxes into vectors and adds them to the token embeddings of their corresponding text as input. 3. Insert Box Embedding (Lu et al., 2024a): Similarly encoding bounding boxes into vectors, it differs from LayoutLM by inserting the encoded position vectors into the original token embedding sequence. For GPE, we also implement the Sinusoidal version GPE-Sinusoidal for comparison. We select Qwen2-7B as the base model. The results of this comparison are shown in Table-2 and Table-3. It can be observed that GPE demonstrates remarkably strong performance across these document tasks. Under our training setup, the Text box and Insert Box Embedding struggles to function effectively, primarily because they alter the input format of the LLM, hence, necessitating an expensive pre-training stage when applied practically. Besides, these two methods have to process a longer input sequence compared to GPE and the method of Add Box Embedding. Add Box Embedding achieved second-best performance, but it is clearly weaker than the GPE method. We argue that this is mainly because Add Box Embedding effectively changes the original input vector space, requiring the model to adapt to a new modality during training. In contrast, GPE directly reuses the original position embedding space, making it easier for the model to learn. For GPE-Sinusoidal, although it uses different type of position embedding with the base model, it still achieves second-best result. This implies that the grouping manner is a better way to incorporate layout information than the other methods.

## 5.5 ANALYSIS

For the following parts, we use Llama2-7B as the base model if not particularly mentioned.
**Influence of Reading Order.** We investigate the impact of reading order part in GPE on model performance, i.e the design of $m$ in the sequence representation $\mathbf{P} = [m, B]$. We compare several designs for reading order as follows. **W/O**: Solely utilize the positional information of text boxes without imposing any specific reading sequence. **Left2Right**: Follow the natural left-to-right, top-to-bottom order as humans typically read. **XYCut**: Following(Gu et al., 2022), obtain proper reading order for rich layout document. **Random**: Randomly shuffle the order of text boxes to arrange tokens in a random sequence. **Local**: Treat each text box equally in this method, with all tokens within a box encoded sequentially starting from 0, ignoring their spatial relationship with other boxes. Considering that the attention structure also influences how reading order is handled, we select two base models for experiments: GLM and qwen2. The results are shown in Table-4. Based on the results, we observe a general concordance in trends between models employing the GLM and Llama2 architectures. Approaches that omit position encoding or utilize random position encoding exhibit inferior performance. The two methods that excel are Local and Left2Right,

Table 4: Ablation on the design of Reading Order representation.

| Method | Llama2 | | | | GLM | | | |
|---|---|---|---|---|---|---|---|---|
| | SynthTables | Forms | SynthDocs | Newspapers | SynthTables | Forms | SynthDocs | Newspapers |
| W/O | 2.4 | 1.7 | 16.5 | 2.5 | 46.6 | 8.6 | 39.9 | 12.6 |
| Random | 63.9 | 31.8 | 63.0 | 26.7 | 66.0 | 15.9 | 47.0 | 32.3 |
| Local | **77.3** | **41.6** | 70.7 | 28.4 | **71.7** | **32.2** | 54.1 | 29.3 |
| Left2Right | 76.4 | 38.7 | **72.7** | **35.7** | 69.8 | 21.2 | **56.2** | **40.3** |
| XYCut | 71.8 | 39.8 | 60.2 | 31.3 | 66.9 | 24.9 | 52.5 | 39.4 |

Table 5: Ablation on the representation methods of layout information on BLADE.

| Setting | W/O Rotation | | | W/ Rotation | | |
|---|---|---|---|---|---|---|
| | SynthTables | Slides | Websites | SynthTables | Slides | Websites |
| LeftPoint | 76.0 | 36.9 | 55.1 | 73.6 | 31.5 | 51.6 |
| Rectangular | 76.4 | 38.7 | **60.5** | 74.3 | 38.8 | **71.6** |
| Quadrilateral | **77.1** | **42.9** | 56.4 | **75.1** | **40.7** | 57.4 |

with each demonstrating a contrasting superiority across different datasets. Specifically, Left2Right excels on SynthDocs and Newspapers, while the Local method outperforms on SynthTables and Forms. We think that these results are influenced by the distinct characteristics of the respective domains. SynthDocs and Newspapers predominantly comprise natural language paragraphs, in which the sequential order of words holds significant importance. Although Left2Right may still occasionally disrupt the reading sequence, it largely preserves sequential information, a feature crucial for understanding narrative coherence in these text-heavy contexts. Conversely, SynthTables and Forms are typified by tabular structures where the reading order among fields is less critical. Here, the Local approach proves advantageous as it enables the model to more effectively leverage spatial information, a key aspect in deciphering structured data arrangements common to tables and forms.

**Design of Layout Information.** We focus on the impact of design of $\mathbf{B}$ in $\mathbf{P} = [m, \mathbf{B}]$. We compare three design approaches. **LeftPoint**: Utilize the coordinates of the top-left corner, where $B = [x_0, y_0]$, with $(x_0, y_0)$ representing the position coordinates of the top-left corner. **Rectangular**: Employ the coordinates of both the top-left and bottom-right corners, expressed as $B = [x_0, y_0, x_1, y_1]$, where $x_i, y_i$ denote the position coordinates of either the top-left ($i = 0$) or bottom-right ($i = 1$) corner. **Quadrilateral**: Involve the coordinates of all four corner points, formatted as $B = [x_0, y_0, x_1, y_1, x_2, y_2, x_3, y_3]$, arranged in a clockwise order, with $(x_i, y_i)$ being the coordinates of each corner point. In each configuration, normalization of coordinates is consistently applied. We employed two conditions, with and without visual rotation, to investigate the impact of coordinate representation. Under the rotated condition, we applied random rotations to the entire evaluation set to emulate the effects of real-world shooting angles. The results are depicted in Table-5. Our findings reveal that the LeftPoint method yields the lowest performance across all evaluation benchmarks, although not significantly underperforming compared to other approaches. This suggests that a substantial portion of layout information within documents is indeed encapsulated by the top-left corner point. The Rectangular and Quadrilateral meethods are largely comparable. Notably, we observed in our experiments that the Rectangular approach converges faster. Regardless of the rotational condition, Quadrilateral marginally outperforms Rectangular, though this advantage is not pronounced. These observations imply that, for the majority of document-related tasks, adopting the Rectangular method is sufficiently effective in achieving commendable outcomes.

**Design of Grouping Function** As previously mentioned, the dimension of $\mathbf{P} = [m, \mathbf{B}]$ and the number of heads in the model typically do not align. Consequently, a grouping function is required to map multiple heads to their corresponding position information. In this part, we explore the effects of varying grouping methodologies. Several straightforward Grouping Functions are: **PadZero** First expand the position vector $\mathbf{P}$ by appending zeros till its dimension match the number of attention heads, then establish a one-to-one correspondence between them; **More Reading Order** Map the majority of heads to the reading order, the rest heads to the layout infomation; **More Layout** Map the majority of heads to layout information, the rest heads to reading order. The results are depicted in Table-6. In terms of results, the strategy of allocating more layout information to the group turns

Table 6: Ablation on the selection of group function on BLADE.

| Group Function | SynthTables | Forms | Slides | Websites | SynthDocs | Newspapers |
|---|---|---|---|---|---|---|
| Baseline | 24.5 | 11.1 | 18.9 | 23.8 | 72.5 | 31.3 |
| PadZero | 2.8 | 10.7 | 1.4 | 2.0 | 11.8 | 1.6 |
| More Reading Order | 76.3 | 33.4 | 50.7 | 65.6 | **74.0** | 35.7 |
| More Layout | **76.4** | **38.7** | **56.2** | **70.5** | 72.7 | **35.7** |

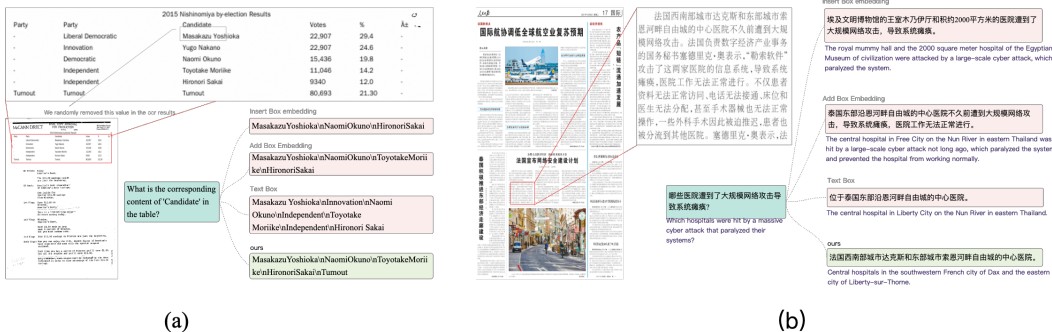

(a)                                                        (b)

Figure 4: Qualitative results on BLADE from Forms and Newspapers. (a) The question requires to list the whole column corresponding to Candidate, only GPE outputs the correct answer. (b) To correct answer the question, model needs to focus on the correct paragraph. Only GPE achieves this.

out to be more advantageous, performing better across nearly all evaluation sets. Coupled with the conclusions in Table-4, Reading Order information is also identified as indispensable.

**Qualitative Results.** We selected two samples from BLADE, and the output results from GPE and other methods are depicted in Figure-4. In the example from Forms, where a complete response to the ''Candidate'' column is required. Other methods either provide additional irrelevant information or miss out on details, whereas only GPE accurately furnishes the answer. In the Newspapers instance, comprehension of the content within a specified area is necessary to answer the question, and again, it is solely GPE that provides the correct response. These two examples illustrate that, with the aid of GPE, LLMs can precisely locate the relevant sections, disregarding distractions from unrelated information, a feat unachievable without the assistance of layout information.

# 6 LIMITATION

While GPE shows promise, several areas remain unexplored. First, although GPE is applicable to any task requiring multidimensional positional encoding, this paper focuses only on document-centric tasks. Graph-structured data could also benefit from similar grouping strategies. Second, most methods, including ours, face a decline in inherent capability, possibly due to structural changes or the quality of document-specific training data. Our approach minimizes alterations to the base model, and we believe future iterations of GPE will further address these challenges.

# 7 CONCLUSION

We introduce Group Position Embedding (GPE), which enhances LLMs' ability to understand multidimensional positional information, particularly in document-related tasks. GPE allows large models to comprehend complex document layouts by injecting independent positional information into grouped attention heads without altering the model architecture or input format. This method outperforms similar approaches. To evaluate large models' understanding of complex documents, we developed BLADE, a challenging benchmark that emphasizes layout information in both data sources and question design. Extensive validation on mainstream document tasks and BLADE demonstrates the effectiveness of our approach.

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

# A APPENDIX

## A.1 DATASETS DESCRIPTION

The training set in this work is composed of the following several parts.

**LayoutLLM-aware SFT Data**(Luo et al., 2024) The layout-aware SFT data for LayoutLLM is generated by GPT-3.5 Turbo and converted from existing text-based Machine Reading Comprehension (MRC) datasets.

**ChartQA Dataset**(Masry et al., 2022) This dataset encompasses 9,600 manually crafted questions along with 23,100 questions generated from manually written chart summaries, aiming to address complex problems involving visual and logical reasoning.

**InfographicVQA Dataset**(Mathew et al., 2022) Designed for the Visual Question Answering (VQA) task on infographics, this dataset comprises around 10,000 images, with an average of about 10 questions per image, each question associated with approximately 4 to 5 answers. Infographics are graphical representations of information, data, or knowledge designed to make complex information more comprehensible, including various types such as charts, flowcharts, and maps.

**DocVQA Dataset**(Mathew et al., 2021) Specialized for document visual question answering tasks, this dataset focuses on textual and graphic elements within documents, assessing machine learning models' performance in understanding and answering questions related to document content. It includes roughly 11,000 images and 40,000 questions.

**VisualMRC Dataset**(Tanaka et al., 2021) Given a question and a document image, this dataset requires machines to read and comprehend the text within the image and answer in natural language. Compared to existing VQA datasets that involve image text, VisualMRC emphasizes nurturing natural language understanding and generation abilities more. It consists of over 30,000 question-answer pairs abstracted from more than 10,000 document images sourced from diverse web domains.

**FUNSD Dataset**(Jaume et al., 2019) Comprising 199 fully annotated real scanned forms with 9,707 semantic entities and 31,485 words, this dataset organizes tables by semantic entity lists. Each entity is identified uniquely, tagged (as question, answer, title, or other), bounded by a box, accompanied by a list of relations to other entities, and a word list. The dataset is split into 149 training and 50 testing samples.

**SROIE Dataset**(Huang et al., 2019) This dataset contains 626 receipts for training and 347 for testing. Each receipt is organized as a list of bounding-boxed text lines. Four types of entities are labeled on each receipt: company, date, address, and total.

**CORD Dataset**(Park et al., 2019) Created for post-OCR parsing, this is the first public dataset containing 11,000 receipt images from Indonesian shops and restaurants. Collected through crowd-sourcing, each image was initially annotated using a web-based tool and then reviewed for accuracy and adherence to annotation guidelines. To avoid unintentional disclosure of sensitive personal data, sensitive information like credit card numbers or full names was blurred in the final receipt dataset.

In addition to the document-related public datasets mentioned above, we have also gathered extra data for fine-tuning LLM instructions. This additional data includes BELLE-CN (3.5 million entries)(BELLEGroup, 2023), FireFly (1.1 million entries)(Yang, 2023), OpenOrca(Yang, 2023), and ShareGPT (90k entries from public). We start by combining these instructions and then randomly select a portion of them to mix with the document data. Through experimentation, we have found that a mixing ratio of document data to text instruction data at 2:1 achieves a well-balanced effect.

## A.2 DETAILS OF BLADE

**Overview**. BLADE comprises six scenarios, SynthTables, Forms, Slides, Websites, SynthDocs, and Newspapers, totaling 3,007 documents and 7,924 question-answer pairs. The specific data volume for each scenario is illustrated in Table-7. The construction of BLADE involves both manual creation and leveraging large language models. The images for Forms are sourced from the (Nan et al., 2022), and those for Newspapers come from the (Cheng et al., 2023). The images for Slides and Websites are obtained through web crawling. The corpus for SynthTables originates from entity names and values extracted from Chinese entity extraction datasets, including (Li et al., 2020),(Tianchi, 2021).

The corpus for SynthDocs is derived from public reading comprehension datasets, including (He et al., 2017),(Cui et al., 2018),(Rajpurkar et al., 2018),(Wang et al., 2020).

| Year | Series | Role | Genre |
|------|--------|------|-------|
| 2013 | Dirt Cheap | Ron Furick | Comedy |
| 2012 | Any Questions for Ben? | Brand Manager | Comedy |
| 2011 | It's Awkward (short) | Hero | Comedy |
| 2006 | Alex and Alexa (short) | Alex | Drama |

**Scheme 1:** Ask for all values corresponding to a header A.

**Q:** What is the corresponding content of 'Genre' in the Table?

**A:** Comedy\nComedy\nComedy\nDrama

**Scheme 2:** Ask for the value of header B when the value of header A is x.

**Q:** When the 'Year' column is '2011', what is the corresponding value in the 'Role'?

**A:** Hero

**Scheme 3:** Ask for the content of the nth row.

**Q:** "In the table, excluding the header row, what is the content of the 2nd row?"

**A:** 2012\nAny Questions for Ben?\nBrand Manager\nComedy

**Scheme 4:** Ask for the value at the nth row and mth column or the mth column and nth row.

**Q:** "What is the 2nd 'Role'?" or "What is the 2nd value corresponding to 'Role'?"

**A:** Brand Manager

Figure 5: Types of questions from SynthTables.

**Data construction.** For the data scenarios of Forms, Slides, and Websites, the predominant method employed was manual annotation, where annotators would select question and answer pairs based on image information, followed by a manual screening process to filter out challenging question-answer pairs. In the case of Newspapers and SynthDocs, given their characteristic of extensive text blocks, a different approach was adopted, utilizing the power of large language models for generation. Specifically, this involved first identifying a key segment within the text, then tasking the large model with constructing question-answer pairs based on that key segment using predefined templates. This way, question-answer pairs that met the required criteria are synthesized.

The distinction between Newspapers and SynthDocs lies in the former's need for an initial manual selection of a text segment from the document, whereas SynthDocs, being synthetically generated, allows for an automated process of segment selection, which is then fed into the large model's pipeline for generating answer pairs. This is because a given key segment text is required in advance for LLM to generate QA pairs. Newspapers are typically documents with complex layouts, making it difficult to obtain the correct reading order through simple OCR sorting. Therefore, we first have annotators to select one key segment, usually a small paragraph, from which the reading order within this small paragraph can be obtained through simple OCR sorting. SynthDocs uses synthetic documents, for which the reading order of the text is already known, thus eliminating the need for manual annotation. The generation pipeline of SynthDocs is shown in Figure-6.

For SynthTables, we also designed an automated pipeline to generate table-question-answer pairs. Specifically, we begin by constructing a set of synthetic tables based on the collected entity data. Following this, SQL is employed to generate a series of question templates on these tables. The questions can be categorized into 4 types, shown in Figure-5. Lastly, we execute the question templates to obtain answers. Through this process, we get a substantial number of table-question-answer pairs.

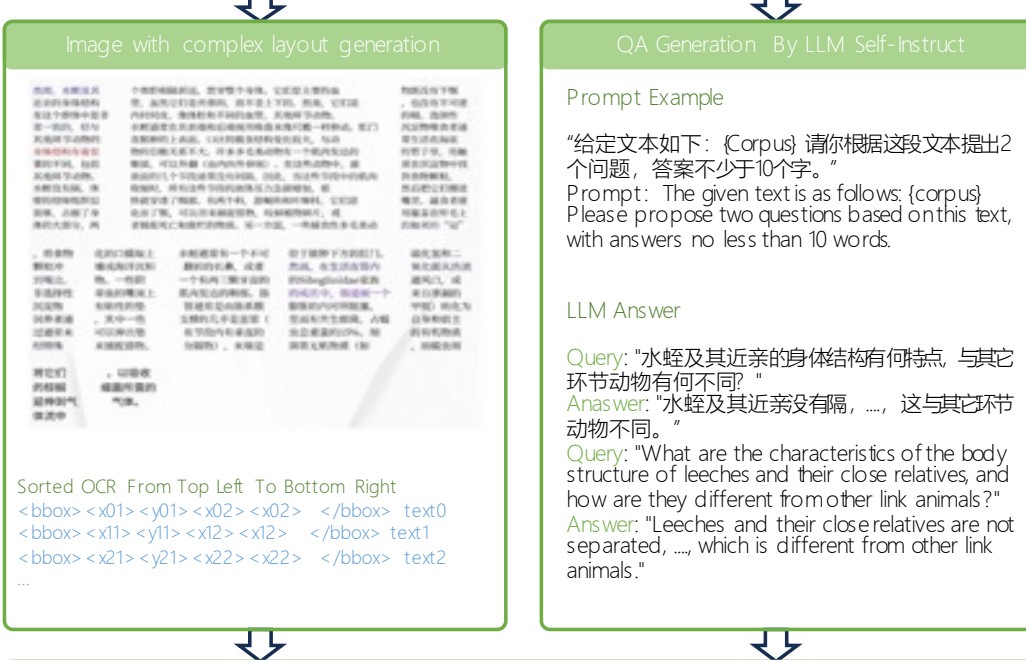

Figure 6: The data generation flowchart of SynthDocs.

Table 7: Statistics of BLADE.

| params | SynthTables | Forms | Slides | Websites | SynthDocs | Newspapers |
|--------|-------------|-------|--------|----------|-----------|------------|
| #docs | 1110 | 162 | 111 | 174 | 1000 | 450 |
| #qas | 2287 | 296 | 518 | 732 | 2000 | 2091 |

These synthetic tables are subsequently rendered into images, with the detailed procedure outlined in Figure-7.

**Quality Verification.** For the SynthDocs data, given that the QA pairs are generated by a LLM, there may be instances of hallucination. To ensure the quality of these generated questions, we utilize LLMs for a screening process during this phase. The quality check primarily focuses on assessing the relevance between the generated questions and answers in relation to the provided text

Figure 7: The data generation flowchart of SynthTables.

content, as well as determining whether any hallucinations have occurred. The specific prompt used for quality screening is as follows:

> **Prompt for quality verification**
>
> Given a text: {text}, there is a question and its corresponding answer, question: {question}, answer: {answer}. Please make judgments on the following aspects:
> 1. Can the question be answered solely based on the content of the material? Please only respond with "Yes" or "No".
> 2. Is the answer given based on the provided text? Please only respond with "Yes" or "No".
> 3. Is the answer correct? Please only respond with "Yes" or "No".
> Return in the format: """{{ 1: "Yes/No", 2: "Yes/No", 3: "Yes/No" }}"""

### A.3 ANALYSIS OF COMMON DOCUMENT BENCHMARKS

In this section, we discuss the characteristics of several commonly used document evaluation benchmarks, specifically two Document Visual Question Answering (VQA) datasets, DocVQA and VisualMRC, as well as three Visual Information Extraction (VIE) datasets, FUNSD, CORD, and SROIE. Based on our previous experiments, the majority of these evaluation tasks fail to assess a model's capability to understand layout information. A robust text large language model, such as qwen2, can generally achieve high scores on these tasks. Our proposed method, GPE, offers limited improvement on top of such a model, and in the case of VisualMRC, including our approach, various methods we implemented to augment layout information actually led to performance degradation, which we attribute to potential impairment of the model's text comprehension abilities.

To fully comprehend the implications behind the numerical variations across these five benchmark datasets, we first examine from the sample level the demands they place on model capabilities. As illustrated in Figure-8 and Figure-9, DocVQA encompasses a variety of photographed documents, where we find that while some questions may necessitate the integration of layout information, the majority can be answered without it, relying solely on the understanding of text paragraphs, as exemplified in samples Figure-8(a). VisualMRC is a collection of various multimedia documents, where accurately answering questions does not require layout information. FUNSD consists of form-like data, similar to DocVQA, yet still presents instances where questions and answers are so straightforward that layout understanding is unnecessary for correct responses. CORD and SROIE involve diverse receipt types, with questions predominantly focused on recognizing entities within receipts.

Evidently, intuitively analyzing these five benchmark sets reveals that except for a minority of questions in DocVQA and FUNSD that might benefit from layout comprehension, the vast majority of tasks, in principle, can be readily tackled with sufficiently powerful text understanding capabilities. Particularly, VisualMRC is fundamentally a reading comprehension task based on large segments of natural text, assessing a model's ability to extract and summarize information from paragraphs. This underscores the motivation behind introducing BLADE, to provide a more nuanced evaluation that goes beyond simple text comprehension and truly examines a model's capacity to understand the layout and structure inherent in document data.

### A.4 TYPE OF POSITION EMBEDDING

In this section, we investigate the impact of the difference between the positional encoding used by GPE and the model's native positional encoding on the performance of downstream tasks. In previous experiments, we implemented two types of encoding: GPE-RoPE and GPE-Sinusoidal. While GPE-Sinusoidal also brings improvements in document tasks, it does not match the performance boost achieved when GPE-RoPE employs the same encoding scheme as the model. Here, we experiment with altering the base scale of GPE-RoPE and observe its performance on BLADE. The results, summarized in Table-8, show that a scale of 1e6 is what the model originally utilizes.

We observe that, on benchmarks akin to reading comprehension question-answering tasks such as Newspapers and SynthDocs, settings close to the original scale yield better performance. For several other categories of tasks, scales approximately around the base scale, like 1e5 and 1e7, exhibit comparable performance. However, significantly deviating from this range, as seen with 1e4, leads to noticeable declines across all tasks. Our experiments have consistently revealed that the design of GPE has differential effects on two types of tasks: those involving extensive natural paragraphs, which emphasize understanding natural language, and those filled with tabular data, which empha-

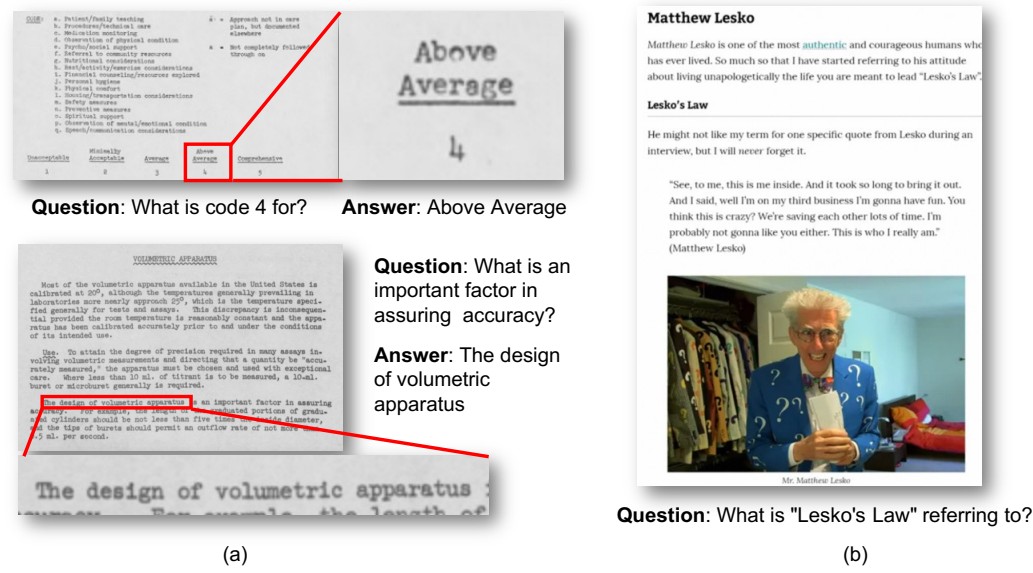

Figure 8: Samples from DocVQA and VisualMRC. (a) DocVQA comprises a variety of photographed documents. Some questions may necessitate the integration of layout information (the upper case), the majority can be answered without it (the bottom case). (b) VisualMRC is a collection of various multimedia documents. Most of the designed question does not need the layout information.

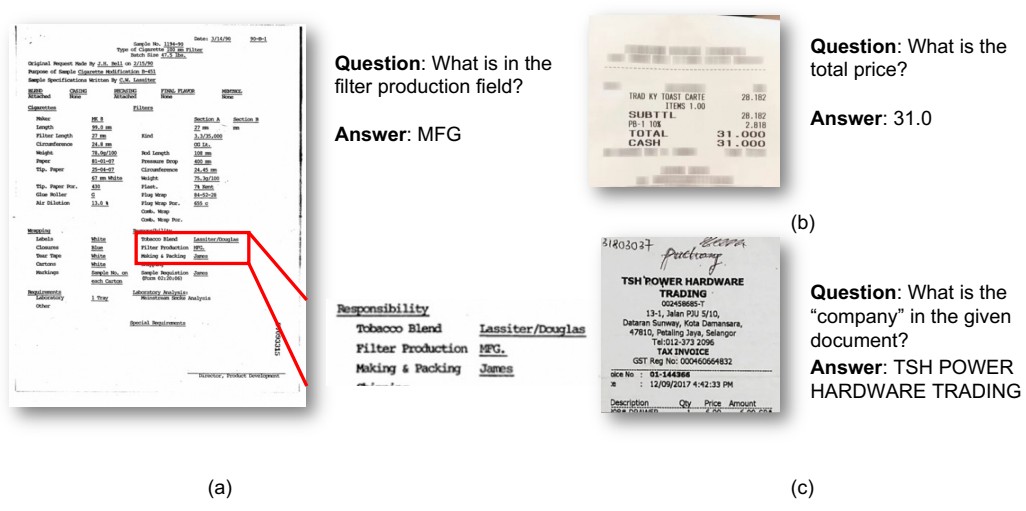

Figure 9: Samples from FUND, CORD and SROIE. (a) FUNSD consists of form-like data. Some questions may need layout information, others not. (b) CORD involves diverse receipt types. Most questions are designed like entity recognition. Layout information is unimportant for accomplishing this task. (c) SROIE is similar to CORD.

Table 8: Ablation on type of position embedding. Qwen2-7B is selected as the base model.

| setting | SynthTables | Forms | Slides | Websites | SynthDocs | Newspapers |
|---|---|---|---|---|---|---|
| Baseline | 50.3 | 33.0 | 42.2 | 46.1 | 53.3 | 43.8 |
| GPE-Sinusodal | 67.2 | 45.6 | 48.3 | 66.5 | 55.2 | 48.7 |
| GPE-RoPE with base scale 1e4 | 77.4 | 54.4 | 59.8 | 79.1 | 58.9 | 48.1 |
| GPE-RoPE with base scale 1e5 | **80.9** | **60.2** | 63.4 | 75.3 | 61.4 | 49.7 |
| GPE-RoPE with base scale 1e6 | 77.3 | 57.2 | **64.5** | **79.3** | **62.2** | **52.2** |
| GPE-RoPE with base scale 1e7 | 77.1 | 58.7 | 63.0 | 76.6 | 61.6 | 49.0 |

size understanding layout relationships. This suite of experimental outcomes suggests that GPE can be optimized for these two types of tasks.

In this work, we adopt a uniform encoding scheme across all heads by default. Exploring more suitable encoding designs separately for the reading sequence component and the coordinate box component, aiming for an even more advantageous GPE configuration, is left as a direction for future research.

## A.5 EFFICIENT TUNING USING LoRA

In previous experiments, we employed full parameter training for our model. Under the setting of full parameter updates, GPE outperforms comparable methods with significantly less data, achieving superior results within a mere few tuning steps. Nonetheless, conducting full parameter updates on large models remains highly resource-intensive. Currently, there exist lightweight training approaches in industry that are based on LoRA, which aim to conserve GPU memory and expedite training by updating only a portion of the model's parameters. The trade-off is that the final performance of the model may not match that of full parameter training; however, this discrepancy varies across different scenarios. In this section, we integrate the training methodologies of GPE with LoRA and examine the model's performance at different steps on BLADE, as illustrated in Figure-10. Our findings reveal that, compared to full parameter training, the LoRA approach requires a longer training duration to reach an equivalent level across all scenarios. Nevertheless, considering that LoRA training for a 7-billion-parameter model can be executed on a single 80GB A100 GPU, the combination of GPE with LoRA presents an efficiently applicable solution in most practical scenarios.

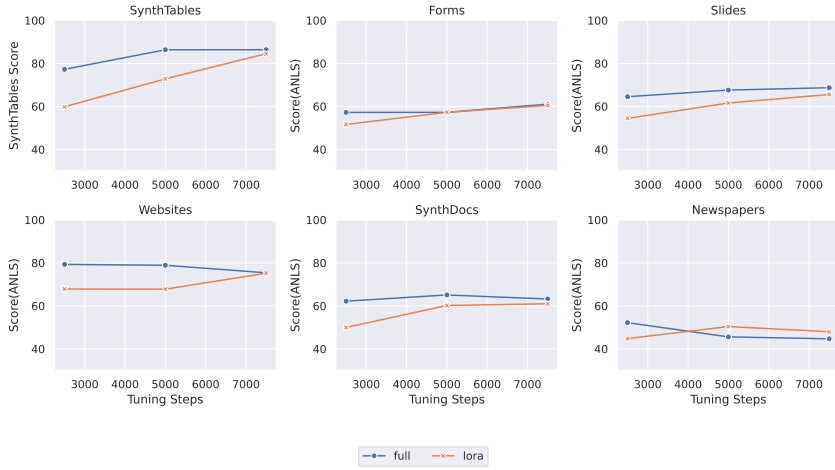

Figure 10: The comparison between full-parameter tuning and LoRA-Tuning on BLADE.

## A.6 INFLUENCE ON BASE MODEL CAPABILITY

We observe that the capabilities of foundation language models are significantly influenced by the document training data they are subjected to. Through experimentation, we have found that this is primarily due to the quality of the document data. Using Qwen-7B as our base model, we mixed textual instruction data with document-type data and adjusted the blending ratio to observe the model's performance on MMLU-Pro(Wang et al., 2024). We discovered that the untrained original Qwen model achieved the best score of 39.55 on MMLU-Pro, but once document-type data was incorporated, the model's text capability was notably impaired. Furthermore, as the proportion of mixed document tasks increased, the decline became more pronounced, aligning closely with the trend observed in GPE. Additionally, under the same data mixing ratio, the performance degradation induced by GPE compared to the original Qwen was not substantial, suggesting that the changes introduced by GPE in position encoding have a minor impact on the model's fundamental abilities. This experiment also revealed that the training data from various document tasks currently integrated can affect the model's inherent capabilities, an aspect that seems to have escaped widespread attention. We believe that this represents a highly promising direction for further research.

Table 9: Influence of data mix ratio on base model's capabilities. The data ratio is organized as textual instruction data vs document data

| Setting | MMLU-Pro | DocVQA |
|---|---|---|
| GPE with mixed data ratio 1:2 | 26.8 | 76.0 |
| GPE with mixed data ratio 1:1 | 28.9 | 72.2 |
| GPE with mixed data ratio 2:1 | 31.2 | 70.8 |
| Qwen with mixed data ratio 1:2 | 27.1 | - |
| Qwen with mixed data ratio 1:1 | 28.4 | - |
| Qwen with mixed data ratio 2:1 | 31.6 | - |
| Qwen-7B-Instruct | 39.6 | - |

## A.7 IMPACT OF SCALING FACTOR

In this section, we explore the influence of the scaling factor $lambda$. We use 1000 in our default implementation. The results are shown in Table-10. It is observed that, with the exception of SynthTables, the variations in performance across the other datasets are relatively minor. However, for SynthTables, higher $lambda$ leads to a significant improvement in performance. We find that this enhancement can be attributed to the dense arrangement of text boxes within SynthTables. In these tables, up to 10 text boxes are often placed in a single row, and their slight positional differences are crucial for determining their column assignments.

By increasing the scaling factor, these subtle positional distinctions become more pronounced, thereby aiding the model in more accurately distinguishing between different cells. This finding underscores the importance of fine-tuning the scaling factor, especially when dealing with datasets that have densely packed and closely positioned elements.

Table 10: The influence of scaling factor.

| $\lambda$ | SynthTables | Forms | Slides | Websites | SynthDocs | Newspapers |
|---|---|---|---|---|---|---|
| 100 | 53.3 | 55.7 | 59.1 | 64.3 | 62.1 | 51.5 |
| 500 | 72.1 | 56.6 | **66.3** | 75.7 | 61.3 | 52.9 |
| 1000(default) | 77.3 | **57.2** | 64.5 | 79.3 | **62.2** | 52.2 |
| 5000 | **86.3** | 56.6 | 66.6 | **79.5** | 61.8 | **53.6** |

A.8  INTUITIVE ANALYSIS OF GPE

The key technique that enhances the effectiveness of GPE is its **head-specific design**. This design ensures that each dimension of layout information is processed by each attention head. The benefits of this approach are akin to those of multi-head attention, as it encourages the model to learn positional relationships from various perspectives. By distributing the layout relations across multiple attention scores, rather than relying on a single score, the model can more effectively capture and utilize different aspects of the layout. Intuitively, some heads might focus on horizontal (left-to-right) relationships, while others concentrate on vertical (up-to-down) relationships.

To intuitively illustrate the impact of GPE on different attention heads, we use heatmaps to visualize the influence of positional information across various attention heads. Specifically, we construct a 2D grid sequence of size 256x256, where each grid point corresponds to a token input into the LLM, and its spatial position is represented by the grid coordinates $[x, y]$. We define the reading order of the grid points using a left-to-right, top-to-bottom scanline sequence. This way, we establish the positional information $\mathbf{P} = [m, x, y]$ for the grid points. We then encode $\mathbf{P}$ using GPE-RoPE with an embedding dimension of 128. We use the grid point at the bottom right corner as the query and observe its attention scores with all other grid points based on their positional information. The results are shown in Figure-11. The visualization clearly demonstrates the contribution of different positional information to the attention scores of different heads.

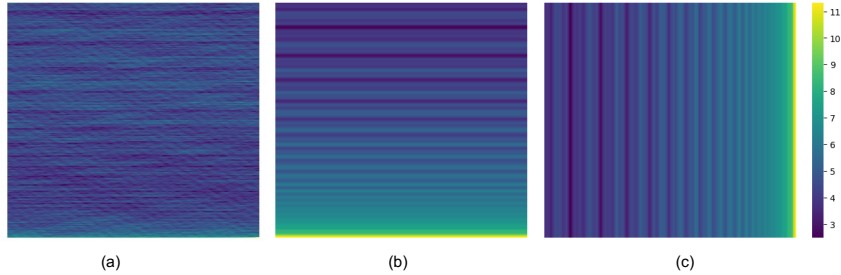

Figure 11: Visualization of attention heatmaps of each head. (a)Attention map with reading order. (b) Attention map with coordinate $x$. (c) Attention map with coordinate $y$.

