# OpenReview forum: "Enhancing Document Understanding with Group Position Embedding: A Novel Approach to Incorporate Layout Information"
_ICLR.cc/2025/Conference — ICLR 2025 Poster_

### Official Review · Reviewer_Ntfi · 2024-10-29

**Soundness:** 3
**Presentation:** 1
**Contribution:** 2
**Rating:** 5
**Confidence:** 4

**Summary:**

This paper proposes a simple yet effective method, Group Position Embedding (GPE), for encoding spatial positional information in LLM-based document understanding tasks. This technique enables LLM models to comprehend document layouts without altering their architectures. Additionally, the authors introduce a new benchmark, BLADE, for evaluating complex document processing.

**Strengths:**

1.This paper introduces a simple layout embedding approach, termed Group Position Embedding (GPE), for enhancing LLM-based document understanding.
2.This paper proposes a new benchmark, BLADE, designed for the evaluation of complex document evaluation.
3.Extensive ablation experiments have been conducted to validate the effectiveness of GPE.

**Weaknesses:**

1. The rationale behind GPE is unclear. The introduction section fails to adequately explain the workings of the group position embedding, instead, it merely highlights the advantages without providing underlying reasons. It is also unclear what the principles and feasibility of the mapping function g_r() are, and the distinction between Gr(k) and Gr(i) on lines 196-197 is also not elucidated. Further explanation is needed on how the n-dimensional spatial position is mapped into different attention heads. The meaning of the hyperparameter scaling factor \lambda and its impact on the approach should also be clarified.
2. This paper is not the first to utilize head-specific layout position embeddings. Previously, LAGaBi [https://aclanthology.org/2023.findings-emnlp.521/] employed diverse Gaussian kernels to encode relative positional information for each attention head.
3.The paper's writing quality is subpar. Inconsistent and non-standard academic expressions are used. For instance, in LLMs, l typically denotes the number of layers rather than dimensions, which should be represented by d or dim. The figures do not clearly demonstrate the mapping mechanism of group position embedding and convey some ambiguity. For instance, Figure 1 shows that only the key-state and value-state encode n-dimensional spatial position information, whereas the query-state receives only 1-dimensional position information, which is confusing.

**Questions:**

Please refer to Weaknesses.

---

> ### Author Response · Authors · 2024-11-17
>
> Q: Why GPE works
>
> A: Please refer to the general response.
>
> Q: Further explanation of group function
>
> A:Following the suggestion, we have revised the paper. The mathematical definition of Group Function is defined in Equation-5.
>
> Q: Effect of $\lambda$
>
> A: The scaling factor lambda does influences the model's performance. We have revised the paper by adding Sec A.7 to address the issue.
>
> We use 1000 in our default implementation. The results are shown following. It is observed that, with the exception of SynthTables, the variations in performance across the other datasets are relatively minor. However, for SynthTables, higher lambda leads to a significant improvement in performance. We find that this enhancement can be attributed to the dense arrangement of text boxes within SynthTables. In these tables, up to 10 text boxes are often placed in a single row, and their slight positional differences are
> crucial for determining their column assignments.
> By increasing the scaling factor, these subtle positional distinctions become more pronounced, thereby aiding the model in more accurately distinguishing between different cells. This finding underscores the importance of fine-tuning the scaling factor, especially when dealing with datasets that have densely packed and closely positioned elements.
> | λ           | SynthTables | Forms | Slides | Websites | SynthDocs | Newspapers |
> |-------------|-------------|-------|--------|----------|-----------|------------|
> | 100         | 53.3        | 55.7  | 59.1   | 64.3     | 62.1      | 51.5       |
> | 500         | 72.1        | 56.6  | 66.3   | 75.7     | 61.3      | 52.9       |
> | 1000(default)| 77.3       | 57.2  | 64.5   | 79.3     | 62.2      | 52.2       |
> | 5000        | 86.3        | 56.6  | 66.6   | 79.5     | 61.8      | 53.6       |
>
> Q: Difference with LAGaBi
>
> A: We have read LAGaBi and believe that there are significant differences between GPE and LAGaBi. Firstly, **the meaning of "head-specific" differs**. In LAGaBi, "head-specific" means that each head has different learnable parameters, but they all use the same positional information, as referenced in Equation 3 of the paper. In contrast, in GPE, "head-specific" means that each head uses positional information from different dimensions of the positional vector. Secondly, **the methods differ**. LAGaBi designs a completely new positional encoding that uses polar coordinates to encode layout information. On the other hand, GPE directly reuses the original model's positional encoding to enhance the model's adaptability. However, we do acknowledge that LAGaBi is indeed a form of layout-aware positional encoding, and we have already included the discussion in the revised paper(refer to the related work).
>
> Q:  l typically denotes the number of layers rather than dimensions, which should be represented by d or dim
>
> A: In line 195, "l" is used to denote the dimmension of the position vector P. "d" has already been used to represent the dimension of the hidden states in Equation-1.
>
> Q: Figure ambiguity. Figure 1 shows that only the **key-state and value-state** encode n-dimensional spatial position information, whereas the **query-state** receives only 1-dimensional position information, which is confusing.
>
> A: We believe the Figure 1 is clear. The **query-state** and **key-state** receives n-dimensional spatial pasition information. **No positional information is applied to value-state**.

---

> > ### Comment · Reviewer_Ntfi · 2024-11-25
> >
> > I'm looking into the revision, still need some time.

---

> > > ### Comment · Reviewer_Ntfi · 2024-11-25
> > >
> > > I really appreciate the detailed explanation, especially the experiments added in Appendix (A.8) for intuitive analysis of GPE, including the attention map visualization, which gives more insights into why GPE works. However, I think the contribution of this work is limited, and the results of the experiment were mixed. I choose to remain with my original score.

---

> ### Author Response · Authors · 2024-11-25
>
> Thank you for taking the time to provide us with your feedback. We would like to clarify the following concerns for the reviewer.
>
> **Limited contribution**
>
> We would like to clarify the contribution here, which has been acknowledged by reviewers. Reviewer XTDc WcWm, hPjd have acknowledged that GPE is a novel method. All reviewers mentioned the dataset contribution, especially WcWm acknowledged that BLADE is important to this field. Three reviewers (WcWm, XTDc, Ntfi) have acknowledged that the extensive experiments that are insightful.
>
> We further summarize the contribution of this work in three aspects
>
> - We propose a **novel head-specific positional encoding approach** that enables LLM to comprehend high-dimensional positional information. This concept is innovative and has not been addressed in previous research.
>
> - Building on the head-specific idea, we devised a **comprehensive method for LLMs to understand document layout information**, supported by **detailed analytical experiments** that showcase various aspects of encoding layout information.
>
> - We also highlight a **limitation in current document-related task evaluation benchmarks**, which primarily rely on LLMs' text comprehension abilities, lacking an assessment of their capacity to understand layout information. To address this gap in evaluation standards, we introduce a new benchmark suite, BLADE, aimed at effectively assessing this aspect of LLM performance.
>
> **Mixed results**
>
> We argue that the statement "The results were mixed" is inaccurate. Through comprehensive experiments, we have demonstrated that GPE outperforms other methods that incorporate layout information. This is clearly evidenced in Tables 2 and 3. Furthermore, as shown in the analysis in Tables 4, 5, 6,8 and 10, GPE can achieve even better performance for specific scenarios . The "mixed results" mentioned by the reviewer likely refer to the differences in the Visual MRC and SROIE datasets in Tables 1 and 2, which are actually as expected. These differences have been analyzed in Sections 5.3, A.3, A.6 and reply to XTDc. The main reason is that both Visual MRC and SROIE primarily measure the model's text capabilities. On datasets that rely more heavily on layout information, such as DocVQA, FUND, and our proposed BLADE, GPE shows a significant advantage.
>
> We sincerely hope that our efforts can address your concerns.

---

### Official Review · Reviewer_WcMm · 2024-11-02

**Soundness:** 3
**Presentation:** 2
**Contribution:** 3
**Rating:** 8
**Confidence:** 5

**Summary:**

The paper makes two main contributions: (1) it uses group position embedding to ensure that different attention heads focus on different views of position, and (2) it introduces a new document AI dataset, BLADE, which highlights the model's ability to handle complex layout information. Experimental results demonstrate improved performance, and the ablation studies are comprehensive.

Updated on 23th Nov 2024: raise the score

**Strengths:**

1. The proposed group position embedding is novel.
2. The proposed BLADE dataset is important to the field.
3. The experiments are extensive, especially the comparison of different approaches to fuse layout information is insightful.

**Weaknesses:**

1. Although the experimental results in Tables 2 and 3 demonstrate the superiority of GPE, the results in Table 1 appear somewhat contradictory to those in Tables 2 and 3, requiring further explanation or experiment.
2. The paper needs a section discussing why GPE is effective in guiding attention heads to focus on different positional views. Including a visualization of attention scores or similar analysis would provide valuable insights, as the improved performance alone is insufficient to fully explain the mechanism.

3. The writing requires improvement, as several paragraphs are vague and difficult to understand. For example:
(1) Lines 248-249: What is the value of lambda? Does GPE discretize all coordinates into integers, allowing us to obtain position embeddings for specific position?
(2) Line 418: In the experiment regarding reading order, I cannot understand the difference between W/O and LOCAL. A concrete example would help illustrate the distinctions between each setting.
(3) Line 473: It would be helpful to introduce the design of the group function where it first appears, in Section 3.2, using a formal mathematical definition. This would clarify the paragraph, as I found it confusing without prior context on the group function.

**Questions:**

See weakness

---

> ### Author Response · Authors · 2024-11-17
>
> Q: Contradictory results.
>
> A: Could you please clarify if the contradiction you are referring to is about the comparison of results between Qwen2 and GPE on VisualMRC? For this part, please refer to the reply to xtdc.
>
> Q: Why GPE works.
>
> A: We have already included a new Section to discuss the intuitive explanation of why GPE works. Please refer to the general response.
>
> Q: Writing details
>
> A: Based on the suggestion. We have revised them in the latest version.
>
> Q: What is the value of lambda? Does GPE discretize all coordinates into integers, allowing us to obtain position embeddings for specific position?
>
>
>
> A: In our implementation, the \lambda is set to 1000. We first discretize each dimension of the coordinate box to a range between 0 and 1000, and then use RoPE for encoding. Additionally, based on reviewer Ntfi's suggestion, we have added experiments to examine the impact of the lambda and find that  lambda does influences the model's performance.  If you are interested, please refer to Sec A.7 in the revised paper.
>
> Q: Difference between W/O and LOCAL
>
> A: We present a example for illustration.
>
> input texts: ["PAGE 01 OF", "MATERIAL SAFETY DATA SHEET"]
>
> input tokens: [[112, 113, 114 ], [115, 116, 117, 118]]  (Note: Tokens in the same list come from one text box)
>
> flattened tokens : [112, 113, 114, 115, 116, 117, 118]
>
> common 1D Position ID:  [0, 1, 2, 3, 4, 5, 6]
>
> W/O 1D Position ID:        [0, 0, 0, 0, 0, 0, 0]
>
> LOCAL 1D Position ID:     [0, 1, 2, 0, 1, 2, 3]
>
> Q: Adjust the position of the Group Function definition.
>
> A:  We have revised the paper. The mathematical definition of Group Function is defined in Equation-5.

---

> > ### Comment · Reviewer_WcMm · 2024-11-23
> > **Thanks**
> >
> > Thanks for the response, since almost all my concerns are addressed, I decided to raise my score to 8.

---

> > > ### Author Response · Authors · 2024-12-03
> > >
> > > We are very pleased that our work has received your recognition. Thank you for your valuable suggestions and feedback and for raising the score.

---

### Official Review · Reviewer_hPjd · 2024-11-02

**Soundness:** 2
**Presentation:** 2
**Contribution:** 2
**Rating:** 6
**Confidence:** 4

**Summary:**

This paper introduces a new positional encoding method to tackle the limitations of current LLMs/MLLMs' lack of layout information for addressing layout-aware document understanding tasks. A dataset named BLADE is designed to mainly focus on measuring the performance on complex-layout aware questions. Various experiments with different model configurations are conducted.

**Strengths:**

1. The paper proposed a new positional encoding to address the limitation of existing LLMs/MLLMs neglecting layout information.
2. A benchmark is proposed specifically focusing on evaluating the performance of various LLMs/MLLMs on document understanding tasks with complex layouts.

**Weaknesses:**

1. Motivation: The limitations of current positional encoding methods adopted by other frameworks are not clearly defined. The research gaps are not clearly defined.
2. The methods are described well with the technical workflow without specific reasons and experiments as to why the positional encoding is working. For example, there is no explanation or citation as to why different position information giving various heads are reasonable for the research aim.
3. The datasets are not clearly described making understanding breakdown categories difficult.
4. The proposed methods are only evaluated on LLM which is expected to see whether it's workable on MLLMs and pretrained document understanding models.
5. More reading order setups should be tried like XY-cut.
6. There are some typos and some possible errors in the paper, like Table 2 SCOIE performance. Some bolded digits in the performance tables are not the highest.

**Questions:**

1. Motivation and Related Work: what is the limitation of current positional encoding methods adopted by pretrained document understanding frameworks and what are the limitations of them?
2. Dataset: is that necessary to have a layout-aware dataset. Only focusing on layout-complex questions may ignoring the performance on other generative tasks.
3. Model and Evaluation: is there any reason you chose those LLMs/MLLMs? It would be better to give more insight analysis for this part by giving more ablation and case studies.
4. Is it possible to show some qualitative analysis of benchmark datasets like CORD, SROIE?

---

> ### Author Response · Authors · 2024-11-17
>
> Q：Motivation and explanation of why GPE works.
>
> A：Please refer to the general response.
>
> Q: Details of BLADE.
>
> A：Please refer to the general response.
>
> Q: Is GPE workable on MLLM?
>
> A: We believe that GPE can also be applied in MLLMs, such as the current scenario of vision-language LLMs (VLMs). In VLMs, visual tokens are flattened and input into the LLM as a one-dimensional sequence, completely losing the two-dimensional information. GPE can help VLMs introduce two-dimensional positional encoding.
>
> Q: More reading order like xy-cut.
> A: Following the suggestion, we have experimented with xy-cut.  We have added the results in the revised paper in Table-4.
>
> The xy-cut method utilizes the spacing in a text library to perform horizontal and vertical divisions, aiming to achieve a better reading order. However, in our experiments, it did not perform well, only showing better results on Forms compared to the left2right approach. After analyzing the data, we found two main reasons for this: 1. SynthTables includes rotation angles, which pose a challenge for the division by xy-cut. 2. The line texts in Docs and Newspapers contain many broken text boxes, a phenomenon that is quite common in real OCR scenarios. The xy-cut method may separate these broken text boxes, leading to a more chaotic reading order.
>
> | Method     |                 |   Llama2        |           |           |                    |    GLM       |           |           |
> |------------|-----------------------|-----------|-----------|-----------|-----------------------|-----------|-----------|-----------|
> |            | SynthTables | Forms  | SynthDocs | Newspapers | SynthTables | Forms  | SynthDocs | Newspapers |
> | W/O        | 2.4         | 1.7    | 16.5      | 2.5       | 46.6       | 8.6    | 39.9      | 12.6       |
> | Random     | 63.9        | 31.8   | 63.0      | 26.7      | 66.0       | 15.9   | 47.0      | 32.3       |
> | Local      | 77.3        | 41.6   | 70.7      | 28.4      | 71.7       | 32.2   | 54.1      | 29.3       |
> | Left2Right | 76.4        | 38.7   | 72.7      | 35.7      | 69.8       | 21.2   | 56.2      | 40.3       |
> | XYCut      | 71.8        | 39.8   | 60.2      | 31.3      | 66.9       | 24.9   | 52.5      | 39.4       |
>
>
> Q:Typos
>
> A: Thank you for your reminder. We have corrected this issue in the newly submitted paper.
>
> Q: Is that necessary to have a layout-aware dataset. Only focusing on layout-complex questions may ignoring the performance on other generative tasks.
>
> A:  A layout-aware dataset is necessary. The purpose of this dataset is to remove semantic influences as much as possible and solely measure the ability of large models to utilize layout information. Currently, in publicly available datasets such as FUNDS and SROIE, a significant number of question-answer pairs can be correctly answered by relying solely on the semantic understanding capabilities of large models, without the need to utilize layout information. Our dataset serves as a complement to the existing public datasets. As shown in Table 1, we not only evaluate the model's ability to utilize layout information but also assess other datasets to evaluate the comprehensive capabilities of the model.
>
> Q: Why choose those LLMs?
>
> A: We chose the ChatGLM-6B, Llama2-7B, and Qwen2-7B models for the following reasons:
>
> (1) **Structural Diversity**: ChatGLM uses prefix attention, where any two tokens in the prefix sequence can see each other. In contrast, both Llama and Qwen adopt casual attention, meaning the model can only access information before the current moment. However, Llama2-7B does not use GQA (Grouped Query Attention), while Qwen2-7B does. These three structures almost represent the common LLMs. Table 1 demonstrates that applying GPE to these three types of LLMs all results in a noticeable improvement, proving the generality of our method.
>
> (2) **Variance in Textual Capabilities**:  Based on Table 1, it is evident that the textual capabilities of ChatGLM-6B, Llama2-7B, and Qwen2-7B increase progressively. Correspondingly, the metrics of our method applied to these three models also show an increasing trend. This indicates that the stronger the pure text performance of a model, the better its performance in document scenarios after applying GPE. As LLMs become more powerful, GPE, which is a low-cost training method for embedding layout information into LLMs, can quickly leverage the robust capabilities of the latest LLMs to achieve superior performance in scenarios requiring layout information.
>
> Q:Qualitative analysis of CORD, SROIE
>
> A: Qualitative analysis is presented in Figure 4 in BLADE. For, CORD and SROIE, the characteristics have been analyzed in Sec A.3 and Figure 8.

---

> > ### Author Response · Authors · 2024-11-25
> >
> > Dear Reviewer  hPjd,
> >
> > We have meticulously addressed your comment with comprehensive responses during the rebuttal phase. At present, we have not yet received new comments from you. We are looking forward to your valuable feedback and insights, and grateful for your effort throughout this process.
> >
> > Best regards,
> >
> > Submission 612 authors

---

> > ### Comment · Reviewer_hPjd · 2024-11-28
> >
> > Thanks for the detailed explanation of my concerns. I think the majority of the problems are answered well. Especially the motivation of this paper. I'm happy to increase my mark from 5 to 6.

---

> > > ### Author Response · Authors · 2024-12-03
> > >
> > > Thank you for taking the time to provide us with your thoughtful and constructive feedback and for raising your score.  We greatly appreciate your valuable support and recognition.

---

### Official Review · Reviewer_XTDc · 2024-11-04

**Soundness:** 3
**Presentation:** 3
**Contribution:** 3
**Rating:** 6
**Confidence:** 2

**Summary:**

This paper introduces Group Position Embedding (GPE), a method to enhance layout comprehension in Large Language Models (LLMs) without requiring architectural changes or extra pre-training. By grouping attention heads with distinct positional embeddings, GPE effectively encodes layout information for document tasks. Tested on five standard benchmarks and the challenging BLADE benchmark, GPE shows significant improvement in document understanding over existing methods.

**Strengths:**

(1) interesting task

(2) new method

(3) extensive experiments

**Weaknesses:**

(1) Missing important details for dataset construction: Any critiria to manual and filter challenging question-answer pairs for the data scenarios of Forms, Slides, and Websites? Why is Newspapers constructed with an initial manual selection whereas SynthDocs being synthetically generated? For SynthDocs whose answers are synthetically generated, any filtering measure to ensure the data quality?

(2) Missing the intuitive motivation of the proposed method: Although this paper verified the effectiveness of the proposed method compared with other approaches on Layout-aware Position Embeddings through the empirical experiments, it lacks of more intuitive explanations on the advantages of the proposed method, which can provide more insights for the follow-up work.

(3) Missing the clarification of the experiment results: According to  the experiment results in Table 1, vanilla Qwen2-7B  outperforms GPE-Qwen2-7B on VisualMRC, which needs more clarifications.

**Questions:**

Please refer to the weaknesses

---

> ### Author Response · Authors · 2024-11-17
>
> Thank you for your valuable feedback. We believe that addressing these details is very helpful for improving our work. We have revised the paper according to your suggestions.
>
> Q：Critiria for manually filter challenging question-answer pairs.
>
> A：The overall standard is to minimize the likelihood that LLMs can infer the correct answer solely based on simply sorted OCR text or semantic information. A QA pair is considered simple if the question text and answer text are placed in natural reading order after simply sorting the OCR text from left to right and up to down. So, in reality,  annotators are required to select samples and question pairs that have spatial interference. For example, in Forms, annotators would select samples with as many rows and columns as possible or select samples that contain rotations. In QA selections, they would choose texts in cells that contain line breaks. In Slides, they would select QA pairs where there are changes in font size or cases where the question and answer are spatially misaligned.
>
> Q: Any filtering measure to ensure SynthDocs' quality?
>
> A: Yes, during the construction of SynthDocs, we used an LLM to filter the generated answer pairs. We have added these details, including the process and the prompt, to the revised paper; please refer to Section A.2.
>
> Q: Why is Newspapers constructed with an initial manual selection whereas SynthDocs being synthetically generated?
>
> A：To generate question-answer pairs using LLMs, a given key segment is required in advance. Newspapers are typically documents with complex layouts, making it difficult to obtain the correct reading order through simple OCR sorting. Therefore, we first have annotators to select one key segment, usually a small paragraph, from which the reading order within this small paragraph can be obtained through simple sorting. SynthDocs uses synthetic documents, for which the reading order of the text is already known, thus eliminating the need for manual annotation.
>
> Q: Missing Intuitive Motivation.
>
> A：Please refer to the general response.
>
> Q: Clarification for results between Qwen2-7B and GPE-Qwen2-7B.
>
> A: We believe that on VisualMRC, the original Qwen2-7B outperforming GPE-Qwen2-7B is as expected. Firstly, **GPE does indeed affect the model's text comprehension ability**, although very slightly. Referring to Sec A.6, the setting with GPE shows a slight decline in text capability (MMLUPro) compared to the original Qwen2-7B.  Secondly, **VisualMRC primarily measures the model's text understanding ability**, as referenced in Sec A.3 and Figure 7. Based on the above two points, we believe that the differences in VisualMRC are as expected.
> In lines 932-935, we actually analyzed this aspect.
>
> Moreover, the gap between GPE and the vallina qwen2-7B is indeed very small. The instability in the calculation of the CIDEr metric results in discrepancies in the scores that are larger than the actual performance differences.
> For example, the following question are both correctly answered by GPE and vallina qwen2-7B. But the CIDEr differs greatly
>
> Question: Do Senior Software Engineer and Senior Cybersecurity Engineer both get the best jobs?
>
> answer: ['Yes, they do.']
>
> qwen2 prediction: Yes, they do. (CIDEr 750.0)
>
> GPE  prediction: Yes.                   (CIDEr 92.3)
>
> It is shown that CIDEr differs greatly in these two answers.
>
> We have revised the paper according to the suggestion, mentioning this point in the main text rather than just in the appendix.

---

> > ### Comment · Reviewer_XTDc · 2024-12-03
> > **Reply to the authors**
> >
> > Thanks for the detailed response from the authors. I think this could address most of my concerns. I have raised my score from 5 to 6.

---

> > > ### Author Response · Authors · 2024-12-03
> > >
> > > Thank you for your insightful and positive feedback, and for raising the score. We greatly appreciate your valuable support and recognition.

---

> ### Author Response · Authors · 2024-11-25
>
> Dear Reviewer XTDc,
>
> We have meticulously addressed your comment with comprehensive responses during the rebuttal phase. At present, we have not yet received new comments from you. We are looking forward to your valuable feedback and insights, and grateful for your effort throughout this process.
>
> Best regards,
>
> Submission 612 authors

---

### Author Response · Authors · 2024-11-27
**Awaiting Reviewer Feedback**

Dear Area Chairs,

We have meticulously addressed each reviewer's comment with comprehensive responses during the rebuttal phase. At present, we have not yet received any responses from 2 reviewers (reviewer hPjd and XTDc). We are looking forward to receiving valuable feedback and insights from them, and appreciate your support throughout this process very much.

Best regards,

Submission 612 authors

---

### Public Comment · ~Yi_Zhang104 · 2024-12-01
**Comments and Concerns Regarding the Paper**

Dear Authors and Reviewers,

We would like to bring to your attention the following points:

1. In Table 1, we noticed that the performance of DocLayLLM and DocLLM on the VIE row appears to be identical. **This should be an error.**

2. We observed that DocLLM employs the F1 metric for the VIE task, which does not align with the ANLS metric mentioned in Table 1. **Is such a comparison appropriate or fair?**

3. LayoutLLM did not utilize the DocVQA, VisualMRC, FUNSD, CORD, or SROIE datasets during training, whereas this paper includes these datasets in the training process. **Could the authors comment on whether this comparison remains fair?** Furthermore, DocLayLLM and LayTextLLM provide experimental results under different settings. **Could the authors confirm whether the comparison made in your paper is based on the zero-shot setting, the VQA setting, or the all-setting?** It would be helpful to understand whether the comparison is fair under these different configurations.

Thanks for this work, and we appreciate your attention to these points. We look forward to your responses.

---

> ### Author Response · Authors · 2024-12-01
>
> Dear Yi Zhang
>
> We are glad to see that our work has caught your attention. We greatly appreciate your interest in our work and your valuable feedback.
>
> **Fair Comparison**
>
> **The comparison in Table 1 is not entirely fair, which has been mentioned in Lines 354-356** .
> >Considering that these methods are based on different base models, trained with varying datasets and strategies, and some without publicly available model weights, the comparison is not entirely equitable.
>
> This is precisely our motivation for the subsequent comparison between Table-2 and Table-3. The unfair comparison in Table-1 does not sufficiently demonstrate the advantages of GPE as an encoding method for enhancing layout information over similar encoding methods, nor does it prove that GPE improves the model's understanding of layout information.
>
> Under these circumstances, we conducted a fair comparison using the exact same setting (Table-2). However, this still does not demonstrate GPE's ability to utilize layout information effectively. To further investigate, we introduced BLADE, a benchmark focused on evaluating a model's understanding of complex layout information, and performed a fair comparison in Table-3. Please refer to Section 5.4 for more details.
>
> Additionally, are you interested in whether using GPE in LLMs can achieve SOTA performance? If so, we hope you will keep an eye on the open-source plans for our work. Recently, we have trained GPE-modified models on larger-scale corpora and instructions, which will support multiple languages and multiple tasks. We will also be using various mainstream LLMs that are currently publicly available within the community.
>
> **Metric of KIE**
>
> Yes, we found that other methods, apart from DocLLM, use the ANLS metric, so we followed most of the work using ANLS. We also provide a comparison of the F1-score of our method with DocLLM here.
>
> | method  | FUND  | CORD   | SROIE  |
> |-------|-------|-------|-------|
> | DocLLM | 51.8 | 67.4 | 91.9 |
> | GPE | 86.8 | 90.1 | 93.9 |
>
> However, as we pointed out in the previous paragraph, such a comparison cannot actually prove the effectiveness of GPE. The introduction of GPE is intended to enhance the understanding ability of LLMs for complex layout information, and we note that these evaluation sets hardly reflect this perspective (Sec A.3). We hope that our proposed BLADE dataset can bring new benchmarks to this field.
>
> **Setting**
>
> In Table-1, we used a non-zero-shot setting, since the training split of these datasets were used during tuning（Sec 5.1). We noticed that you mentioned DocLayLLM and LayTextLLM using two settings, and the ones cited in Table 1 are from the non-zero-shot setting.  LayoutLLM only reports zero-shot setting. Its results were cited as a reference.
>
> **Reference Error**
>
> Thanks for your reminding. There are some citation errors in the results of this DocLayLLM. We will fix this problem.
>
> We hope our response can answer your questions, and we thank you once again for your interest in our work.

---

### Author Response · Authors · 2024-12-01
**Response for general concern**

## Revised Paper and Future work
We have revised and re-uploaded the paper. The key changes are highlited in yellow.  The revised part mainly includes
- Comparison with LAGaBi in related work
- Grouping function definition
- A new Section(A.7) for the impact of scaling factor and experiments.
- Adding experiment of XY-cut for comparison
- More details of BLADE.
- A new Section(A.8) for intuitive analysis of GPE including the attention map visualization.
- Some typos

Future work of open source plan includes:
- BLADE and its evaluation scripts.
- Code of GPE as well as its training and evaluation
- Release model weights based on SOTA LLMs with GPE using large scale training data

## Motivation and Ituitive Analysis of GPE
**Motivation**
Our objective is to integrate layout information into LLMs. Given the limitations of existing methods, we have identified three key characteristics that our approach should embody:
1. **Unambiguous Layout Representation**: The representation of the layout should not result in any loss of information.
2. **No Additional Vocabulary**: Introducing new vocabulary or modalities would complicate the training process.
3. **No Additional Tokens**: Adding extra tokens would disrupt the original text sequence, making the LLM hard to understand the original text.

The GPE is designed to meet all the properties. GPE satisfies the first property by ensuring that each dimension of layout information is processed individually by different heads.  In contrast, the ``Add Box Embedding'' used in LayoutLM just adds the embeddings of each layout dimension, which brings ambiguity, e.g. the representations of [x1, x2] and [x2,x1] are identical. GPE satisfies property 2 by reusing the original position embeddings of LLM. As it is not a new modality or new vocabulary, the model is easy to adapt. GPE does not change the input sequence thus satisfying property 3.

**Explanation of why GPE is effective**
The key technique that enhances the effectiveness of GPE is its **head-specific design**. This design ensures that each dimension of layout information is processed by each attention head. The benefits of this approach are akin to those of multi-head attention, as it encourages the model to learn positional relationships from various perspectives. By distributing the layout relations across multiple attention scores, rather than relying on a single score, the model can more effectively capture and utilize different aspects of the layout. Intuitively, some heads might focus on horizontal (left-to-right) relationships, while others concentrate on vertical (up-to-down) relationships.

**Visualization of Attention Map**
Please refer to the newly added Section A.8 in the revised paper.

## Details of BLADE
We have added more details to BLADE, which includes two parts: (1) an explanation of the differences in the construction details between Newspapers and SynthDocs; (2) additional information on the quality filtering process for SynthDocs.

---

### Meta-Review · Area_Chair_nNus · 2024-12-17

**Metareview:**

The authors present a novel approach, Group Position Embedding (GPE), to enhance document understanding by enabling attention heads to focus on different positional views. The authors also propose a new benchmark, BLADE, for complex document processing, offers a valuable contribution to the field.

To further strengthen the paper, I recommend incorporating a more comprehensive discussion of related work, particularly in the area of layout encoding for language models. While the authors have considered recent advancements in LLMs and LMMs, exploring earlier work, such as ROPE: Reading Order Equivariant Positional Encoding for Graph-based Document Information Extraction (ACL 2021), can provide valuable insights and historical context.

**Additional Comments On Reviewer Discussion:**

The initial reviewer feedback highlighted concerns regarding the motivation and intuition behind the design, as well as the dataset construction and evaluation methodology. The authors have made significant efforts to address these concerns, providing more detailed explanations and justifications.

---

### Decision · Program_Chairs · 2025-01-22

Accept (Poster)